# A Hierarchical Spatial Transformer for Massive Point Samples in Continuous Space

**Wenchong He**     **Zhe Jiang**∗     **Tingsong Xiao**     **Zelin Xu**     **Shigang Chen**
Department of Computer & Information Science & Engineering
University of Florida
{whe2, zhe.jiang, xiaotingsong, zelin.xu, sgchen}@ufl.edu

**Ronald Fick**     **Miles Medina**     **Christine Angelini**
Center for Coastal Solutions
University of Florida
{rfick, miles.medina}@ufl.edu, christine.angelini@essie.ufl.edu

## Abstract

Transformers are widely used deep learning architectures. Existing transformers are mostly designed for sequences (texts or time series), images or videos, and graphs. This paper proposes a novel transformer model for massive (up to a million) point samples in continuous space. Such data are ubiquitous in environment sciences (e.g., sensor observations), numerical simulations (e.g., particle-laden flow, astrophysics), and location-based services (e.g., POIs and trajectories). However, designing a transformer for massive spatial points is non-trivial due to several challenges, including implicit long-range and multi-scale dependency on irregular points in continuous space, a non-uniform point distribution, the potential high computational costs of calculating all-pair attention across massive points, and the risks of over-confident predictions due to varying point density. To address these challenges, we propose a new hierarchical spatial transformer model, which includes multi-resolution representation learning within a quad-tree hierarchy and efficient spatial attention via coarse approximation. We also design an uncertainty quantification branch to estimate prediction confidence related to input feature noise and point sparsity. We provide a theoretical analysis of computational time complexity and memory costs. Extensive experiments on both real-world and synthetic datasets show that our method outperforms multiple baselines in prediction accuracy and our model can scale up to one million points on one NVIDIA A100 GPU. The code is available at https://github.com/spatialdatasciencegroup/HST.

## 1  Introduction

Transformers are widely used deep learning architectures. Existing transformers are largely designed for sequences (texts or time series), images or videos, and graphs [1, 2, 3, 4, 5, 6, 7]. This paper proposes a novel transformer model for massive (up to a million) points in continuous space. Given a set of point samples in continuous space with explanatory features and target response variables, the problem is to learn the spatial latent representation of point samples and to infer the target variable at any new point location.

Learning transformers for continuous-space points has broad applications. In environmental sciences, researchers are interested in fusing remote sensing spectra with in-situ sensor observations at irregular

---

∗Contact author

37th Conference on Neural Information Processing Systems (NeurIPS 2023).

sample locations to monitor coastal water quality and air quality [8, 9, 10]. In scientific computing, researchers learn a neural network surrogate to speed up numerical simulations of particle-laden flow [11] or astrophysics [12]. For example, in cohesive sediment transport modeling, a transformer surrogate can predict the force and torque of a large number of particles dispersed in the fluid and simulate the transport of suspended sediment [13, 14]. In location-based service, people are interested in analyzing massive spatial point data (e.g., POIs, trajectories) to recommend new locations [15, 16].

However, the problem poses several technical challenges. First, implicit long-range and multi-scale dependency exists on irregular points in continuous space. For example, in coastal water quality monitoring, algae blooms in different areas are interrelated following ocean currents and sea surface wind (e.g., Lagrangian particle tracking). Second, point samples can be non-uniformly distributed with varying densities. Some areas can be covered with sufficient point samples while others may have only very sparse samples. Third, because of the varying sample density as well as feature noise and ambiguity, model inference at different locations may exhibit a different degree of confidence. Ignoring this risks over-confident predictions. Finally, learning complex spatial dependency (e.g., all-pair self-attention) across massive (millions) points has a high computational cost.

To address these challenges, we propose a new hierarchical spatial transformer model, which includes multi-resolution representation learning within a quad-tree hierarchy and efficient spatial attention via coarse approximation. We also design an uncertainty quantification branch to estimate prediction confidence related to input feature noise and point sparsity. We provide a theoretical analysis of computational time complexity and memory costs. Extensive experiments on both real-world and synthetic datasets show that our method outperforms multiple baselines in prediction accuracy and our model can scale up to one million points on one NVIDIA A100 GPU.

## 2 Problem Statement

A *spatial point sample* is a data sample drawn from 2D continuous space, denoted as $\mathbf{o_i} = (\mathbf{x}(\mathbf{s}_i), y(\mathbf{s}_i), \mathbf{s}_i)$, where $1 \leq i \leq n$, $\mathbf{s}_i \in \mathbb{R}^2$ is 2D spatial location coordinates (e.g., latitude and longitude), $\mathbf{x}(\mathbf{s}_i) \in \mathbb{R}^{m \times 1}$ is a vector of $m$ non-spatial explanatory features, and $y(\mathbf{s}_i)$ is a target response variable ($y(\mathbf{s}_i) \in \mathbb{R}$ for regression, $y(\mathbf{s}_i) \in \{0, 1\}$ for binary classification). For example, in water quality monitoring, a spatial point sample consists of non-spatial explanatory features from spectral bands of an Earth imagery pixel, the spatial location of that pixel in longitude and latitude, and ground truth water quality level (e.g., algae count) at that location.

We aim to learn the target variable as a continuous-space function $y : \mathbb{R}^2 \rightarrow \mathbb{R}$ or $\{0, 1\}$. Given a set of point sample observations $\mathcal{O} = \{(\mathbf{x}(\mathbf{s}_i), y(\mathbf{s}_i), \mathbf{s}_i)\}_{i=1}^n$, where $\mathbf{s}_i$ is irregularly sampled in 2D, our model learns the contiguous function $y$ that can be evaluated at any new spatial points $\hat{\mathbf{s}} \notin \{\mathbf{s}_i\}_{i=1}^n$. We define our model as learning the mapping from the observation samples to the continuous target variable function $y(\hat{\mathbf{s}}) = f_\theta(\mathcal{O}, \mathbf{x}(\hat{\mathbf{s}}), \hat{\mathbf{s}})$. Thus, we formulate our problem as follows.

**Input:** Multiple training instances $\mathcal{D} = \{\mathcal{O}_j\}_{j=1}^L$ (sets of irregular points in continuous 2D space).
**Output:** A spatial transformer model $f : \{y(\hat{\mathbf{s}}), u(\hat{\mathbf{s}})\} = f(\mathbf{x}(\hat{\mathbf{s}}), \hat{\mathbf{s}}, \mathcal{O}_j)$ for any $j \in [1, ..., L]$, where $\hat{\mathbf{s}}$ is any new sample location for inference, and $\mathbf{x}(\hat{\mathbf{s}})$ and $y(\hat{\mathbf{s}})$ are the explanatory features and output target variable for the new sample, respectively, and $u(\hat{\mathbf{s}})$ is the uncertainty score corresponding to the prediction $y(\hat{\mathbf{s}})$.
**Objective:** Minimize prediction errors and maximize the uncertainty quantification performance.
**Constraint:** There exists implicit multi-scale and long-range spatial dependency structure between point samples in continuous space.

Note that to supervise model training, we can construct the training instances by removing a single point sample from each set $\mathcal{O}_j$ as the new sample location and use its target variable as the ground truth.

## 3 Related Work

• **Transformer models:** Attention-based transformers are widely used deep learning architecture for sequential data (e.g., texts, time series) [1], images or videos [5, 6], graphs [7]. One main advantage of transformers is the capability of capturing the long-range dependency between samples. One major computational bottleneck is the quadratic costs associated with the all-pair self-attention. Various

techniques have been developed to address this bottleneck. For sequential data, sparsity-based methods [17, 18] and low-rank-based methods [19, 20, 21] have been developed. Sparsity-based methods leverage various attention patterns, such as local attention, dilated window attention, or cross-partition attention to reduce computation costs. For low-rank-based methods, they assume the attention matrix can be factorized into low-rank matrices. For vision transformers, patch-based [22, 23] or axial-based [24, 25] have been proposed to improve computational efficiency. Similarly, for graph data, sampling-based [26, 27, 28] and spectral-based [29, 30] attention mechanisms were proposed to reduce computation complexity. However, these techniques require an explicit graph structure with fixed topology. To the best of our knowledge, existing transformer models cannot be applied to massive point samples in continuous space.

• **Numerical Operator learning in continuous space:** Neural operator learning aims to train a neural network surrogate as the solver of a family of partial differential equation (PDE) instances [31]. The surrogate takes initial or boundary conditions and predicts the solution function. Existing surrogate models include deep convolutional neural networks [2], graph neural operators [3, 32], Fourier neural operators [33, 34], DeepONet [31], NodeFormer [26, 35] and vision transformers [36, 37, 38]. However, existing methods are mostly designed for regular grid or fixed graph node topology and thus cannot be applied to irregular spatial points. There are several methods for irregular spatial points in continuous space through implicit neural representation [39, 40, 41], or fixed-graph transformation [42], but their neural networks are limited by only taking each sample's individual spatial coordinates without explicitly capturing spatial dependency.

• **Deep learning for spatial data:** Extensive research exists on deep learning for spatial data. Deep convolutional neural networks (CNNs) [43, 44] are often used for regular grids (e.g., satellite images, and global climate models) [45, 46], and graph neural networks (GNNs) [47] are used for irregular grids (e.g., meshes with irregular boundaries) [48, 49, 50] or spatial networks (e.g., river or road networks) [51, 52, 53]. However, CNNs and GNNs only capture local spatial dependency without long-range interactions. In recent years, the transformer architecture [1, 54] has been widely used for spatial representation learning with long-range dependency, but existing transformer models are often designed for regular grids (images, videos) [5, 6, 23, 24, 25, 22] and thus cannot be directly applied to irregular point samples in continuous space.

## 4 Approach

This section introduces our proposed **hierarchical spatial transformer (HST)** model. Figure 1 shows the overall architecture with an encoder and a decoder. The encoder learns a multi-resolution representation of points via spatial pooling within a quad-tree hierarchy (quadtree pooling) and conducts efficient hierarchical spatial attention via point coarsening. The intuition is to approximate the representation of faraway key points by the representation of coarse quadtree cells. The decoder makes inferences at a new point location by traversing the quadtree and conducting cross-attention from this point to all other points. The decoder also contains an uncertainty quantification (UQ) branch to estimate the confidence of model prediction related to feature noise and point sparsity. Note that our model differs from existing tree-based transformers [55, 56] for images or videos, as these methods require a regular grid and cannot be directly applied to irregular points in continuous space.

### 4.1 Multi-resolution representation learning within a quadtree hierarchy

Learning a multi-resolution latent representation of point samples is non-trivial due to the non-uniform (irregular) distribution of points in continuous space (instead of a regular grid in images [22]). To address this challenge, we propose to use a quadtree to establish a multi-scale hierarchy, a continuous spatial positional encoding for point representation, and a spatial pooling within a quadtree hierarchy to learn multi-resolution representations.

### 4.1.1 Spatial representation of individual points (quadtree external nodes)

**Continuous spatial positional encoding**: A positional encoding is a continuous functional mapping $\phi : \mathbb{R}^2 \to \mathbb{R}^{\frac{d}{2}}$ from the 2D continuous space to a $\frac{d}{2}$-dimensional encoded vector space. The encoding function needs to allow a potentially infinite number of possible locations in continuous space and the similarity between the positional encodings of two points should reflect their spatial proximity, i.e.,

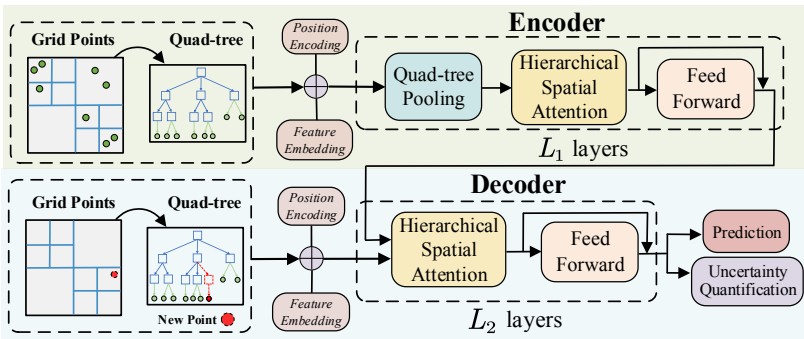

Figure 1: The overall architecture of our hierarchical spatial transformer model.

nearby samples tend to have a higher dot product similarity in their positional encoding. Common positional encodings based on discrete index numbers for sequence data are insufficient. We propose to use a multi-dimensional continuous space position encoding [57] as follows,

$$\phi(\boldsymbol{s}) \approx [\cos(\Omega_1 \boldsymbol{s}), \sin(\Omega_1 \boldsymbol{s}), ..., \cos(\Omega_{\frac{d}{2}} \boldsymbol{s}), \sin(\Omega_{\frac{d}{2}} \boldsymbol{s})], \tag{1}$$

where $d$ is the encoding dimension, $\Omega_i \sim \mathcal{N}(\mathbf{0}, \Sigma)$ is a 1 by 2 projection matrix following an i.i.d. Gaussian distribution with a standard deviation $\sigma$. The advantage of this encoding is that it satisfies the following property, $< \phi(\boldsymbol{s}_1), \phi(\boldsymbol{s}_2) > \approx k(||\boldsymbol{s}_1 - \boldsymbol{s}_2||) = \exp\{-(\boldsymbol{s}_1 - \boldsymbol{s}_2))^T \Sigma^{-1} (\boldsymbol{s}_1 - \boldsymbol{s}_2)\}$. Here the hyperparameter $\Sigma$ controls the spatial kernel bandwidth. Next, we use $\phi(\boldsymbol{o}_i)$ to denote the positional encoding $\phi(\boldsymbol{s}_i)$ for consistency.

**Spatial representation:** We propose an initial spatial representation of individual points by concatenating its continuous-space positional encoding and a non-spatial feature embedding, i.e.,

$$\mathbf{h}(\boldsymbol{o}_i) = [\boldsymbol{\psi}(\boldsymbol{o}_i); \boldsymbol{\phi}(\boldsymbol{o}_i)] \tag{2}$$

where $\phi(\boldsymbol{o}_i)$ is a positional encoding, $\boldsymbol{\psi}(\boldsymbol{o}_i) = \mathbf{W} \cdot [\mathbf{x}_i; y]$ is the non-spatial embedding, $\mathbf{W} \in \mathbb{R}^{\frac{d}{2} \times (m+1)}$ is the embedding parameter matrices, and $d$ is the dimension of concatenated representation. We denote the representation of all point samples (external quadtree nodes) in a matrix $\mathbf{H}_o = [\boldsymbol{h}(\boldsymbol{o}_1), ... \boldsymbol{h}(\boldsymbol{o}_n)]^T \in \mathbb{R}^{n \times d}$, where each row is the representation of one point.

### 4.1.2 Spatial representation of coarse cells (quadtree internal nodes)

To learn a multi-resolution representation of non-uniformly distributed points in continuous space, we propose a spatial pooling operation within a quadtree hierarchy. A **quadtree** is a spatial index structure designed for a large number of points in 2D continuous space. It recursively partitions a rectangular area into four equal cells until the number of points within a cell falls below a maximum threshold. In a quadtree, an internal node at a higher level represents an area at a coarser spatial scale and nodes at different levels provide a multi-resolution representation. Another advantage of using a quadtree is that it can handle non-uniform point distribution. A subarea with denser points will have a deeper tree branch through continued recursive space partitioning.

Formally, given the set of point samples $\mathcal{O} = \{\mathbf{o}_i\}_{i=1}^N$ in continuous space, we construct a quadtree $\mathcal{T}$. The quadtree has two kinds of node sets: an external node set $\mathcal{E}$, which corresponds to the observed spatial point samples, and an internal node set $\mathcal{I}$, which represents the spatial cells. A quadtree has $L$ levels, and all the nodes in level $l$ form a set $\mathcal{R}_l = \{r_1^l, ..., r_{k_l}^l\}$, where $r_j^l$ is the $j$-th node at level $l$, and $k_l$ is the total number of nodes in level $l$. Given one node $r_j^l$, we represent its sibling node set as $\mathcal{S}(r_j^l)$ (nodes on the same hierarchical level under the same parent node), the ancestor node set as $\mathcal{A}(r_j^l)$ (nodes on the path from the node $r_j^l$ to the root). We denote the spatial representation of the node $r_j^l$ as $\mathbf{h}(r_j^l)$ and $l \in \{1, ...L\}$ and $j \in \{1, ..., k_l\}$.

For example, in Figure 2(a), there are 11 input point samples with denser point distribution in the upper left corner and the lower right corner. Assuming a maximum leaf node size of two samples, the

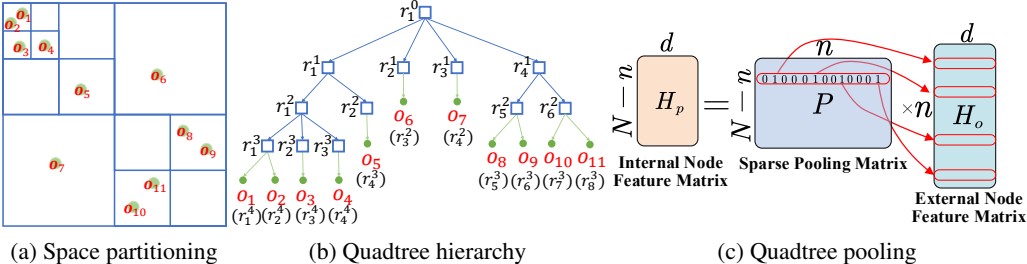

(a) Space partitioning      (b) Quadtree hierarchy      (c) Quadtree pooling

Figure 2: An example of a quadtree and pooling operation.

corresponding quadtree is shown in Figure 2(b), which has 12 internal nodes (blue) and 11 external nodes (green). There are five different levels starting with level 0 (the root node). Level 1 has four internal nodes, i.e., $\mathcal{R}_1 = \{r_1^1, ..., r_4^1\}$. For instance, $r_1^1$ corresponds to the largest quad cell in the upper left corner. It has two non-empty children nodes at level 2, one of which $r_1^2$ is an internal node and the other of which $r_2^2$ is a leaf node linked to an external node $r_4^3$ (also expressed as $o_5$). The sibling set $\mathcal{S}(r_1^2)$ is $\{r_2^2\}$ (the other two sibling nodes are empty cells and are thus ignored). The set of ancestors $\mathcal{A}(r_1^2)$ is $\{r_1^1, r^0\}$.

Assume the total number of quadtree nodes is $N$, including $n$ leaf nodes (point samples) and $N - n$ internal nodes (coarse cells). We can compute the representation of each internal node by an average pooling of the representation of its children within the quadtree hierarchy (i.e., **quadtree pooling**).

Formally, the spatial representation $\mathbf{H}_p$ for all quadtree internal nodes can be computed by sparse matrix multiplication, as shown in Equation 3, where $\mathbf{H}_o \in \mathbb{R}^{n \times d}$ is the representation matrix of $n$ point samples (external nodes), $\mathbf{H}_p \in \mathbb{R}^{(N-n) \times d}$ is the pooled representation matrix of $N - n$ internal nodes, and $\mathbf{P} \in \mathbb{R}^{(N-n) \times n}$ is a sparse pooling matrix. Each row of $\mathbf{P}$ is normalized and its non-zero values indicate all the corresponding external nodes (point samples) under an internal node. The computational structure is shown by Figure 2(c). We concatenate the internal node feature $\mathbf{H}_p$ and external nodes feature $\mathbf{H}_o$ to form a representation matrix $\mathbf{H} \in \mathbb{R}^{N \times d}$ for all quadtree nodes.

$$\mathbf{H}_p = \text{QuadtreePooling}(\mathbf{H}_o) = \mathbf{P}\mathbf{H}_o, \tag{3}$$

### 4.2 Efficient Hierarchical Spatial Attention

The goal of the spatial attention layer is to model the implicit long-range spatial dependency between all sample points. Computing all-pair self-attention across massive points (e.g., a million) is computationally prohibitive due to high time and memory costs. To reduce the computational bottleneck, we propose an efficient hierarchical spatial attention operation based on a coarse approximation of key points. Specifically, instead of computing the attention weight from a query point $o_i$ to all other points as keys, we only compute the weight from $o_i$ to a selective subset of quadtree nodes. Our intuition is that for key points that are far away from the query point $o_i$, we can use coarse cells (nodes) to approximate them, and the further away those points are, the coarser cells (upper-level nodes) we can use to approximate them.

Formally, for each query point (external node), we define its **key node set** as $\mathcal{K}$, which includes the point itself, its own siblings (points) as well as the siblings of its ancestors. That is, $\mathcal{K}(o_i) = \{o_i\} \cup \mathcal{S}(o_i) \cup \{\mathcal{S}(r) | r \in \mathcal{A}(o_i)\}$. For example, in Figure 3, the key set of the external node $o_1$ (also noted as $r_1^4$) is $\{o_1, o_2, r_2^3, r_3^3, r_2^2, r_2^1, r_3^1, r_4^1\}$.

In this way, we reduce the number of attention weight calculations from all 11 points to 8 quadtree nodes. Particularly, we use one quadtree node $r_4^1$ to approximate all the four points within it ($o_8$ to $o_{11}$) since they are far away from $o_1$. Based on the definition of a key set, the spatial attention operator can be expressed as Equation 4 below, where $h_i$ is the output representation for $o_i$, $\mathcal{K}_i$ is the key node set of $o_i$, $q_i$ is the query vector of point $o_i$, $k_j$ and $v_j$ are the key vector and value vector of

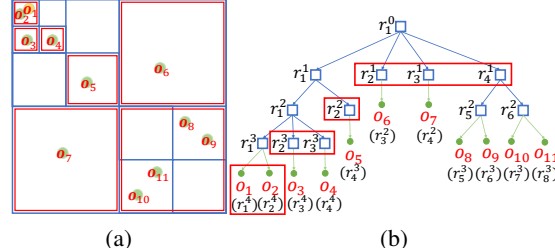

(a)      (b)

Figure 3: An example of a quadtree (a) and the selective key node set of $o_1$ in red boxes (b).

the attended node $r_j$, respectively, and $d$ is the latent dimension of these vectors.

$$\boldsymbol{h}_i = \sum_{j \in \mathcal{K}_i} \frac{\exp(\boldsymbol{q}_i \boldsymbol{k}_j^T / \sqrt{d}) \boldsymbol{v}_j}{\sum_{j \in \mathcal{K}_i} \exp(\boldsymbol{q}_i \boldsymbol{k}_j^T / \sqrt{d})} \tag{4}$$

We can express the spatial attention operator in matrix and tensor notations. Assume the spatial representation of all quadtree nodes from the prior layer is $\mathbf{H} \in \mathbb{R}^{N \times d}$ and the representation of external nodes (point samples) as $\mathbf{H}_o$. The query matrix of all point samples can be computed by an embedding with learnable parameter matrix $\mathbf{W}_q$, i.e., $\mathbf{Q}_o = \mathbf{W}_q \mathbf{H}_o$. For simplicity, we denote the query matrix $\mathbf{Q}_o$ as $\mathbf{Q}$. For each query point $\mathbf{o}_i$, its keys are a subset of quadtree nodes. We denote their embedded key vectors in a matrix $\mathbf{K}_i \in \mathbb{R}^{d \times |\mathcal{K}_i|}$. If we concatenate the key matrices for all queries, we get a 3D tensor $\hat{\mathbf{K}} \in \mathbb{R}^{d \times |\mathcal{K}_i| \times n}$. Similarly, the corresponding value matrices can be concatenated into a 3D tensor $\hat{\mathbf{V}} \in \mathbb{R}^{d \times |\mathcal{K}_i| \times n}$. The construction of 3D tensors can be implemented by the torch.gather() API in Pytorch. The corresponding cross-attention weights can be calculated by matrix and tensor multiplications, as illustrated in Figure 4. We can see the difference between the proposed hierarchical spatial attention and the default all-pair attention. In the all-pair self-attention (Figure 4 top), we would have to compute the dot product attention for all point entries, i.e., $\mathbf{Q}_o^T \cdot \mathbf{K}_o$, whose size is $n \times n$ and can become too large (e.g., $n = 1,000,000$ for a million points). In our spatial attention layer, we conduct a sparse self-attention, i.e., only computing the attention weight of a sample point to its key set. In other words, the corresponding key matrix for $\mathbf{o}_i$ is a vertical slicing of $\hat{\mathbf{K}}$, which can be denoted as $\hat{\mathbf{K}}[:,:,i] \in \mathbb{R}^{d \times |\mathcal{K}_i|}$, where $|\mathcal{K}|$ is the maximum key node set size ($|\mathcal{K}| << n$).

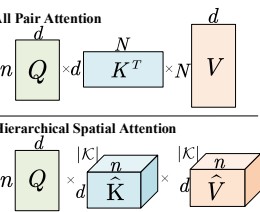

Figure 4: Sparse spatial attention with selective key set.

**Time cost analysis:** The proposed spatial attention computation cost depends on the uniformness of spatial point distribution, which determines how balanced the quadtree is. Assume that there are $n$ points and the maximum number of points in each quadtree node is $M$ (quadtree threshold). We analyze two scenarios. If the quadtree is completely balanced, the tree depth and the key set size for every external node is $O(\log \frac{n}{M})$. Thus total computation cost is $O(n \cdot (M + \log \frac{n}{M}))$. In the worst case, the spatial points are highly nonuniform, then the quadtree structure will be highly imbalanced. The extreme tree depth is $O(\frac{n}{M})$, thus the attention computation is $O(n \cdot (\frac{n}{M} + M))$. However, such a worst-case scenario is unlikely in practice, as it would require samples to be concentrated within a single subgroup of the quadtree at every level. For instance, for water quality monitoring, the sensors are sparsely distributed across a broad area to monitor multiple locations. In practical applications, it takes $O(n \cdot (\log \frac{n}{M} + M))$ complexity. This is validated in our experiments.

**Memory cost analysis:** We analyze the memory costs of the HST model theoretically. Assume the number of input point samples $n$, leaf node size threshold $M$, batch size $B$, and the number of head $h$, and hidden dimension $d$. The memory costs of HST are dominated by the hierarchical spatial attention layer. Its memory cost is $O(B \cdot h \cdot d \cdot n \cdot |\mathcal{K}_i|)$ per layer, where $|\mathcal{K}_i| = \log \frac{n}{M} + M$ for relatively balanced quadtree.

### 4.3 Decoder: inference on a new point with uncertainty quantification

**Model inference (prediction):** Given a test sample $\mathbf{o}_t = (\mathbf{x}_t, \mathbf{s}_t)$, the decoder module predicts $y_t$ based on learned spatial representations of quadtree nodes $\mathbf{H}$. Similar to the encoder, we use cross-attention between the test sample and its corresponding key node set in the quadtree as in Equation 4. As shown in Figure 1, we first conduct a quadtree traversal of the test point until reaching the leaf node and then identify the key node set. Based on that, we can apply a hierarchical spatial cross-attention between the test location and its key node set, followed by a dense layer.

**Uncertainty quantification (UQ):** Due to the varying point density as well as feature noise and ambiguity, the model prediction at a new location may come with different confidence levels. Intuitively, the prediction at a test location surrounded by many input point samples tends to be more confident. Many methods exist in UQ for deep learning (e.g., MC dropout and deep ensemble [58, 59, 60, 61]), but few consider the uncertainty due to varying sample density. The most common

method for UQ in the continuous spatial domain is the Gaussian process (GP) [62, 63]. The UQ of GP is based on Equation 5, where $\mathbf{c}_0 \in \mathbb{R}^{1 \times n}$ is the covariance vector between the test sample and all input point samples, $\mathbf{C} \in \mathbb{R}^{n \times n}$ is the covariance matrix for input samples, and $\sigma_0^2$ is the self-variance. In GP, the covariance $\mathbf{C}$ is computed with a kernel function that reflects the location proximity [64]. Although a GP has good theoretical properties, it is inefficient for massive points due to expensive matrix inverse operation on a large covariance matrix, and it is unable to learn a non-linear representation for samples.

$$\sigma_t^2 = \sigma_0^2 - \mathbf{c}_0^T \mathbf{C}^{-1} \mathbf{c}_0 \tag{5}$$

Our proposed spatial transformer framework can be considered a generalization of a GP model. We use the dot-product cross-attention weights to approximate the covariance vector $\mathbf{c}_0$ between the test location to all points, which reflects the dependency among point sample locations based on their non-linear embeddings, i.e., $\mathbf{c}_0 = \mathbf{Hq}_t$, where $\mathbf{q}_t$ is the embedded query vector for the test sample. However, this idea is insufficient for the approximation of the entire covariance matrix $\mathbf{C}$ across all points, since the inverse computation of a full covariance matrix is very expensive. To overcome this challenge, we propose to directly approximate the precision matrix ($\mathbf{C}^{-1}$) based on an indicator matrix of selective key sets for all queries $\mathbf{S} \in \mathbb{R}^{N \times n}$, in which each column indicates the key node set of a query point (and thus specifies the dependency between a query point to all other quadtree nodes). Thus, we use $\mathbf{S}^T \cdot \mathbf{S}/T_u^2$ to approximate the precision matrix $\mathbf{C}^{-1}$, where $T_u^2$ is a hyper-parameter to be calibrated by independent validation data with calibrated with the Expected Calibration Error (ECE) [65]. The intuition is that the precision matrix reflects the conditional independence structure among point samples (similar to the selective spatial attention in our model). Since $\mathbf{S}^T \cdot \mathbf{S} = \sum_i \mathbf{s}_i \mathbf{s}_i^T$ ($\mathbf{s}_i$ is a column of $\mathbf{S}$), we can see that $\mathbf{S}^T \cdot \mathbf{S}$ is a summation of several sparse block-diagonal matrices. This reflects our assumption in the quadtree that each external node (point sample) is conditionally independent of all other nodes given its key node set. Based on the above approximation, our uncertainty quantification method can be expressed by Equation 6, where $\mathbf{K}_t = \mathbf{SH}$ is the key matrix corresponding to the selective key set of the test sample.

$$\begin{aligned} u_t &= \sigma_0^2 - (\mathbf{q}_t^T \mathbf{H}^T) \mathbf{S}^T \mathbf{S} (\mathbf{Hq}_t)/T_u^2 \\ &= \sigma_0^2 - \mathbf{q}_t^T (\mathbf{SH})^T (\mathbf{SH}) \mathbf{q}_t/T_u^2 \\ &= \sigma_0^2 - (\mathbf{K}_t \mathbf{q}_t)^T (\mathbf{K}_t \mathbf{q}_t)/T_u^2 \end{aligned} \tag{6}$$

Note that our approach shares similarities with the multipole graph neural operator model (MGNO) [32]. MGNO employs a multi-scale low-rank matrix factorization technique to approximate the full kernel matrix across all samples. A key distinction lies in that MGNO uses a neighborhood graph structure to approximate the dependency relationships. In contrast, our approach can capture the long-range interactions among samples in Euclidean space and we provide uncertainty quantification in prediction.

## 5 Experimental Evaluation

The goal is to compare our proposed spatial transformer with baseline methods in prediction accuracy and uncertainty quantification performance. All experiments were conducted on a cluster installed with one NVIDIA A100 GPU (80GB GPU Memory). The candidate methods for comparison are listed below. **The models' hyper-parameters are provided in the supplementary materials.**

**Gaussian Process (GP):** We used the GP model based on spatial location without using the explanatory feature [62]. The prediction variance was used as the uncertainty measure. **Deep Gaussian Process (Deep GP):** We implememted a hybrid Gaussian process neural network [66] with the sample explanatory features and locations. The GP variance is used as the uncertainty. **Spatial graph neural network (Spatial GNN):** We first constructed a spatial graph based on each sample's kNN by spatial distance. Then we trained a GNN model [47]. We used the MC-dropout [58] method to quantify the prediction uncertainty. **Multipole graph neural operator (MGNO):** It belongs to the family of neural operator model [32]. **NodeFormer :** It is an efficient graph transformer model for learning the implicit graph structure [26]. We used the code from its official website. **Galerkin Transformer:** It uses softmax-free attention mechanism and acheieve linearlized transformer [35]. We quantify the prediction uncertainty based on MC-dropout. **Hierarchical Spatial Transformer (HST):** This is our proposed method. We implemented it with PyTorch.

The prediction performance is evaluated with mean square error (MSE) and mean absolute error (MAE). The evaluation of UQ performance is challenging due to the lack of ground truth of uncertainty.

**UQ evaluation metrics**: The quantitative evaluation metrics for UQ performance is Accuracy versus Uncertainty ($AvU$)[67]. We set accuracy uncertainty thresholds $T_{ac}, T_{au}$ to group prediction accuracy and uncertainty into four categories as Table 1 shows. $n_{AC}, n_{AU}, n_{IC}, n_{IU}$ represent the number of samples in the categories AC, AU, IC, IU, respectively. As Equation 7 shows, $AvU$ measures the percentage of two categories AC and IU. A reliable model should provide a higher $AvU$ measure ($AvU \in [0, 1]$). The details on how we choose the thresholds are provided in the supplementary materials.

$$AvU = \frac{n_{AC} + n_{IU}}{n_{AC} + n_{AU} + n_{IC} + n_{IU}} \tag{7}$$

However, $AvU$ is usually biased by the accuracy of the model. Since models tend to have high confidence in accurate predictions. We propose an evaluation metric that evaluates uncertainty performance for accurate and inaccurate prediction separately. Specifically, we computed $AvU_A$ for accurate predictions and $AvU_I$ for inaccurate predictions with the following equations:

$$AvU_A = \frac{n_{AC}}{n_{AC} + n_{AU}}, AvU_I = \frac{n_{IU}}{n_{IC} + n_{IU}} \tag{8}$$

In our evaluation, we computed the harmonic average of $AvU_A$ and $AvU_I$: $AvU = \frac{2 * AvU_A * AvU_I}{AvU_A + AvU_I}$ to penalize the extreme cases instead of the arithmetic average.

Table 1: Accuracy versus Uncertainty (AvU)

| | | Uncertainty | |
|---|---|---|---|
| | | Certain | Uncertain |
| Accuracy | Accurate | Accurate Certain (AC) | Accurate Uncertain (AU) |
| | Inaccurate | Inaccurate Certain (IC) | Inaccurate Uncertain (IU) |

**Dataset description**: We used three real-world datasets, including two water quality datasets collected from the Southwest Florida coastal area, and one sea-surface temperature and one PDE simulation dataset: **Red tide dataset:** The input data are satellite imagery obtained from the MODIS-Aqua sensor [68] and *in-situ* red tide data obtained from Florida Fish and Wildlife's (FWC) HAB Monitoring Database [69]. We have $104, 100$ sensor observations and we use a sliding window with a sequence length 400 to generate 103700 inputs. It is split into training, validation, and test sets with a ratio of $7 : 1 : 2$. **Turbidity dataset:** We used the same satellite imagery features as in the red tide dataset. The ground truth samples measure the turbidity of the coastal water. It contains 13808 sensor observations. **Darcy flow for PDE operator learning**: The Darcy flow dataset [33] contain 100 simulated images with $241 \times 241$ resolution. For each image, we subsample 100 sets of point samples from the original image (each set has 400 nodes). **Sea Surface Temperature:** We used the sea surface temperature dataset of Atlantic ocean [70]. We subsampled 400 point samples from the grid pixels.

Table 2: Comparison of model performance on two real-world datasets and one simulation dataset

| Model | Red tide | | Turbidity | | Darcy flow | |
|---|---|---|---|---|---|---|
| | MSE | MAE | MSE | MAE | MSE | MAE |
| GP | $7.42 \pm 0.25$ | $2.55 \pm 0.18$ | $0.42 \pm 0.02$ | $0.46 \pm 0.03$ | $0.19 \pm 0.03$ | $0.33 \pm 0.04$ |
| Deep GP | $6.23 \pm 0.42$ | $2.32 \pm 0.24$ | $0.35 \pm 0.04$ | $0.42 \pm 0.06$ | $0.18 \pm 0.03$ | $0.31 \pm 0.05$ |
| Spatial GNN | $5.68 \pm 0.07$ | $2.19 \pm 0.04$ | $0.34 \pm 0.02$ | $0.46 \pm 0.03$ | $0.15 \pm 0.02$ | $0.26 \pm 0.04$ |
| All-pair transformer | $5.30 \pm 0.12$ | $1.98 \pm 0.07$ | $0.31 \pm 0.03$ | $\mathbf{0.35 \pm 0.04}$ | $\mathbf{0.10 \pm 0.02}$ | $\mathbf{0.22 \pm 0.03}$ |
| MGNO | $5.41 \pm 0.26$ | $2.04 \pm 0.10$ | $0.32 \pm 0.03$ | $0.38 \pm 0.05$ | $0.12 \pm 0.03$ | $0.27 \pm 0.03$ |
| NodeFormer | $5.34 \pm 0.10$ | $2.05 \pm 0.06$ | $0.32 \pm 0.04$ | $0.38 \pm 0.04$ | $0.16 \pm 0.03$ | $0.29 \pm 0.04$ |
| GalerkinTransformer | $5.44 \pm 0.17$ | $2.11 \pm 0.08$ | $0.34 \pm 0.03$ | $0.42 \pm 0.05$ | $0.11 \pm 0.02$ | $0.25 \pm 0.03$ |
| **HST (Our method)** | $\mathbf{5.25 \pm 0.11}$ | $\mathbf{1.97 \pm 0.07}$ | $\mathbf{0.30 \pm 0.03}$ | $\mathbf{0.35 \pm 0.04}$ | $0.11 \pm 0.02$ | $0.23 \pm 0.03$ |

## 5.1 Comparison on prediction performance

We first compared the overall regression performance between the baseline models and our proposed HST model. The results on two datasets were summarized in Table 2. The first column summarizes the performance of the red tide dataset with MSE and MAE evaluation metrics. We can see that the GP model performed the worst due to the ignorance of the samples' features. The deep GP

model improved the GP performance from $7.42$ to $6.23$ by leveraging the neural network feature representation learning capability as well as considering point sample correlation in both spatial and feature proximity. The spatial GNN model performed well because the model considered local neighbor dependency structure and feature representation simultaneously. However, the spatial GNN model relies on proximity-based graph structure so it ignores the spatial hierarchical structure and long-range dependency. MGNO baseline model performs slightly better than spatial GNN because the multi-level message-passing mechanism captures the long-range dependency structure. Our framework performs better because of the awareness of multi-scale spatial relationships, which is crucial for point samples' inference. Additionally, the performance of all-pair attention is better than the fix-graph GNN model because of the long-range modeling. However, the vanilla transformer models lack the hierarchical attention structure, which may cause an imperfect attention weight score. In contrast, our model performed best with the lowest MAE of $5.25$ because it modeled the interaction among point samples in the continuous multi-scale space efficiently. We can see similar results on the MAE metrics. For the second turbidity dataset, our model consistently performed the best in accuracy. For the PDE operator learning task, the results are shown in Table 2. We can see our proposed framework outperforms baselines except all-pair transformer but our model is more efficient. In summary, compared with recent SOTA graph transformer and neural operator learning methods, our framework performs best due to the capability of learning multi-scale spatial representations among massive point samples. Compared with the all-pair transformer model, our HST reduces the time costs and makes a trade-off between efficiency and spatial granularity when calculating attention between points. More experiment results on the Sea Surface Temperature dataset are provided in the supplementary materials.

**Sensitivity analysis:** We also conducted a sensitivity analysis of our model to various hyper-parameters, including the quadtree leaf node size threshold $M$, spatial position encoding length scale $\sigma$, the number of attention layers, and the embedding dimension on the red-tide dataset. The results are summarized in Figure 5 and the detailed analysis is in the Supplementary material. We can see that our model is generally stable with changes of the hyper-parameters.

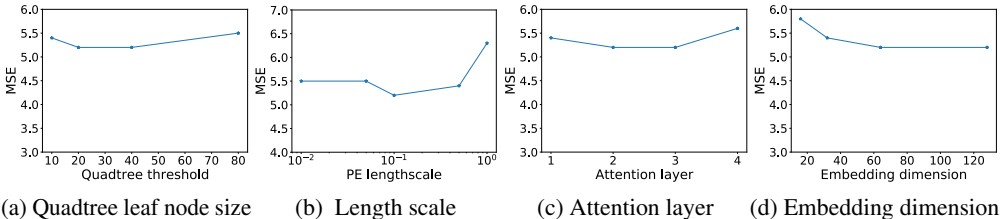

| (a) Quadtree leaf node size | (b) Length scale | (c) Attention layer | (d) Embedding dimension |

Figure 5: Parameters sensitivity analysis.

## 5.2 Comparison on uncertainty quantification performance (UQ)

We use $AvU$ in Equation 7 to evaluate the performance of UQ for our proposed HST model versus baseline methods and the results were summarized in Table 3. The numbers in the table correspond to the number of samples in four categories: $n_{AC}, n_{AU}, n_{IC}, n_{IU}$.

We can see the base GP model had good uncertainty estimation for inaccurate predictions ($65\%$ are uncertain) but for the accurate predictions, the uncertainty estimation tended to be confident, only $14\%$ were certain. This might be due to sparse area prediction being less confident but the long-range spatial correlation or feature similarity can improve prediction in the sparse sample area, but the GP model was unaware of such dependency. The deep GP model improved the accurate prediction confidence based on the feature representation learning but still performed worse than other models and gave a $0.38$ $AvU$ score. The spatial GNN model was

Table 3: Comparison on uncertainty quantification performance on Red tide dataset

| Model | Accurace | Uncertainty | | AvUa/ AvUc | AvU |
|---|---|---|---|---|---|
| | | Certain | Uncertain | | |
| GP | Accurate | 2726 | 9144 | 0.23 | 0.33 |
| | Inaccurate | 3252 | 4919 | 0.60 | |
| Deep GP | | Certain | Uncertain | | |
| | Accurate | 3316 | 8509 | 0.28 | 0.38 |
| | Inaccurate | 3094 | 5122 | 0.62 | |
| Spatial GNN | | Certain | Uncertain | | |
| | Accurate | 3152 | 3949 | 0.44 | 0.40 |
| | Inaccurate | 8355 | 4585 | 0.35 | |
| Galerkin Transformer | | Certain | Uncertain | | |
| | Accurate | 4515 | 4925 | 0.47 | 0.38 |
| | Inaccurate | 7166 | 3435 | 0.32 | |
| **HST** | | Certain | Uncertain | | |
| | Accurate | 4342 | 5302 | 0.46 | 0.54 |
| | Inaccurate | 3874 | 6503 | 0.65 | |

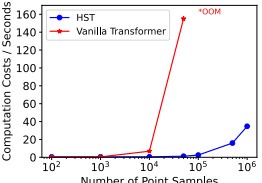 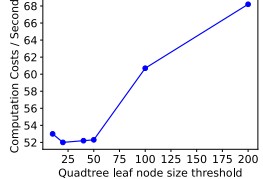 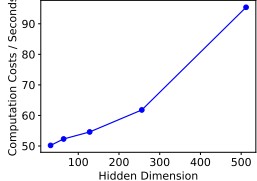

(a) Computation cost per epoch for 64 training samples.

(b) Computation cost per epoch versus leaf node size threshold for 5K training samples.

(c) Computation cost per epoch versus hidden dimension for 5K training samples.

Figure 6: Computation cost analysis

more confident in the accurate prediction, but
the model was over-confident in the inaccurate prediction, thus resulting in a lower $AvU$ score (0.44). In contrast, our uncertainty model improved both the accurate prediction confidence and inaccurate prediction uncertainty, and improved the overall $AvU$ score to 0.49, because it modeled the uncertainty coming from both feature space and sample density simultaneously. The results validate our decoder attention capability in modeling the prediction uncertainty.

## 5.3 Analysis on computation and memory cost

We evaluate the computation costs on a simulation dataset. The point samples' locations are uniformly distributed on the two-dimensional space, and we generate the input feature and target label based on the simulated Gaussian Process. The simulated number of point samples varies from $10^2$ to $10^6$. The training batch size is 1. Other simulation parameters and training hyperparameters are provided in the Appendix. We compare our model with the vanilla all-pair attention transformer on both computation times. The computational time per epoch on the 64 training dataset is shown in Figure 6(a). When the number of point samples increases to 50K, the computation time and memory costs of the vanilla all-pair transformer increase dramatically and become out-of-memory (OOM) when further increasing the number to 100K. However, our model can be scaled to 1M point samples and trained with a reasonable time cost (1 hour). We also analyze the effect of the quadtree leaf node size threshold (the maximum number of point samples in the quadtree leaf) using 5K training samples. The computation time is shown in Figure 6(b). When the leaf node size threshold increase, the computation first decrease and then increase. Because when the quadtree depth decreases from 10 to 25, the quadtree depth decreases and can decrease the total number of query-key pairs. When the threshold increases from 50 to 200, the leaf node point sample will increase the computation. The computational costs with hidden feature dimensions are shown in Figure 6(c). It's observed that the computational costs scale linearly with respect to the hidden feature dimension.

## 6 Conclusion and Future Work

This paper proposes a novel hierarchical spatial transformer model that can model implicit spatial dependency on a large number of point samples in continuous space. To reduce the computational bottleneck of all-pair attention computation, we propose a spatial attention mechanism based on hierarchical spatial representation in a quadtree structure. In order to reflect the varying degrees of confidence, we design an uncertainty quantification branch in the decoder. Evaluations of real-world remote sensing datasets for coastal water quality monitoring show that our method outperforms several baseline methods. In future work, we plan to extend our model to learn surrogate models for physics-simulation data on 3D mesh and explore improving the scalability of our model on multi-GPUs.

## Acknowledgement

This material is based upon work supported by the National Science Foundation (NSF) under Grant No. IIS-2147908, IIS-2207072, CNS-1951974, OAC-2152085, the National Oceanic and

Atmospheric Administration grant NA19NES4320002 (CISESS) at the University of Maryland, and Florida Department of Environmental Protection. Shigang Chen's work is supported in part by the National Health Institute under grant R01 LM014027.

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
