# 7 Supplementary Material

## 7.1 Baseline models and Hyperparameters

• **Gaussian Process (GP):** We trained the GP model based on spatial location without using the explanatory feature [62]. The prior is constant mean and RBF kernel, and we optimized the model parameters by maximum likelihood estimation. The prediction variance was used as the uncertainty measure. • **Deep Gaussian Process (Deep GP):** We implememted a hybrid Gaussian process neural network [66]. The sample explanatory features were fed into a multi-layer perceptron, then the learned latent features and sample spatial locations were fed into a Gaussian process model. The GP variance is used as the uncertainty measure. • **Spatial graph neural network (Spatial GNN):** We first constructed a spatial graph based on each sample's k-nearest-neighbor by spatial distance. Then we trained a GNN model [47] on the constructed graph structure. The model contains two GCN layers. To evaluate the uncertainty of GNN models, we use the MC-dropout [58] method. The method does multiple forward dropout inferences at test time and computes the variance of the prediction as the uncertainty measure. • **Multipole graph neural operator (MGNO):** MGNO [32] belongs to the family of neural operator models that learn the infinite-dimensional function mapping in the continuous space. It contains a multi-level graph neural network to capture the long-range interactions among particles with linear complexity. We use the PyTorch implementation from its official website. • **NodeFormer:** NodeFormer [26] is an efficient graph transformer model for learning the implicit graph structure among samples. It utilizes kernelized random Fourier features for efficient attention learning. • **Galerkin Transformer:** GalerkinTransformer [35] uses softmax-free attention mechanism and acheieve linearlized transformer. The approach derives a Petrov-Galerkin interpretation to approximate the operator. We also quantify the prediction uncertainty based on the MC dropout method. The dropout rate is tuned based on the validation dataset. • **Hierarchical Spatial Transformer (HST):** This is our proposed method. We implemented it with PyTorch. The model architecture and hyperparameters are in the hyperparameter paragraph.

**Model hyper-parameters**: Our HST model used three spatial attention layers in both the encoder and the decoder, the latent representation embedding dimension was 64, and the quadtree leaf node size threshold was 20 by default. For the training process, we used the MSE loss with a decaying learning rate that reduced the learning rate by half if the validation loss did not improve over five epochs (with an initial learning rate of $10^{-4}$ and a minimum rate of $10^{-7}$). We also used early stopping with a patience of 10 epochs and a maximum of 50 epochs. The optimizer was Adam with $\beta_1 = 0.9$ and $\beta_2 = 0.98$. The $L_2$ regularization weight was $10^{-4}$. The batch size was 512.

**Uncertainty evaluation hyper-parameters:** We use an accuracy and uncertainty threshold $T_{ac}$ and $T_{au}$ to group samples' prediction into four categories: accurate certain, accurate uncertainty, inaccurate certain, and inaccurate uncertain. The threshold for accuracy is determined by the mean absolute error (MAE) loss $T_{ac} = \frac{\sum_{i=1}^{n}(y_i - \hat{y})}{n}$, where $n$ is the number of validation samples. The accurate prediction index set is $\mathcal{A} = \{i | |y_i - \hat{y}| < T_{ac}\}$, and inaccurate prediction index set is $\mathcal{I} = \{i | |y_i - \hat{y}| > T_{ac}\}$. We use $T_{au}^A$ and $T_{au}^I$ to represent the uncertainty threshold for accurate and inaccurate prediction, respectively. Then $T_{au}^A = \text{Average}(\mathcal{U}_A)$, where $\mathcal{U}_A = \{u_i | i \in \mathcal{A}\}$. Then $T_{au}^I = \text{Average}(\mathcal{U}_I)$, where $\mathcal{U}_I = \{u_i | i \in \mathcal{I}\}$.

**Dataset description**: We use three real-world datasets, including two water quality datasets collected from the Southwest Florida coastal area, and one sea-surface temperature and one PDE simulation dataset to evaluate our proposed model: • **Red tide dataset** The input data are satellite imagery obtained from the MODIS-Aqua sensor and *in-situ* red tide data obtained from Florida Fish and Wildlife's (FWC) HAB Monitoring Database. We have $104,100$ point samples and it is split into training, validation, and test sets with a ratio of $7 : 1 : 2$. • **Turbidity dataset:** We used the same satellite imagery features as in the red tide dataset. The ground truth samples measure the turbidity of the Southwest Florida coastal water. It contains $13808$ point samples. The dataset is split into train validation and test sets with the same ratio as the red tide dataset. • **Darcy flow for PDE operator learning**: The Darcy flow dataset contain 100 simulated images with 241*241 resolution . For each image, we subsample 100 graph samples from the original image, where each graph contains 400 nodes. We construct a KNN graph based on the node coordinates. For the test dataset, we do inference on the whole grid map, thus the test dataset contains 100 images with 241*241 resolution. • **Sea Surface Temperature:** We used the sea surface temperature dataset of Atlantic ocean [70]. The dataset consists of daily temperature acquisition on $481 \times 781$ pixels from 2006 to 2017. The

Table 4: Comparison on model performance on Sea Surface Temperature dataset

| Model | Sea Surface Tem | |
|---|---|---|
| | MSE | MAE |
| GP | 1.68 | 1.25 |
| Spatial GNN | 1.60 | 1.17 |
| All-pair transformer | **1.52** | **1.15** |
| HST (Our method) | 1.56 | **1.15** |

image is divided into $64 \times 64$ subregions, and we subsampled $400$ point samples from the grid pixels. We use the years from 2016 to 2017 as the test dataset and the years from 2015 and 2016 as the validation dataset.

## 7.2 Comparison on prediction performance

In this section, we present the result for sea surface temperature. The results are shown in Table 4. We can observe that all-pair transformers perform the best and our model performs better than GP and spatial GNN models. For this dataset, we do not consider the temporal aspect of temperature dynamics, and we only focus on the prediction of continuous spatial locations.

## 7.3 Sensitivity analysis

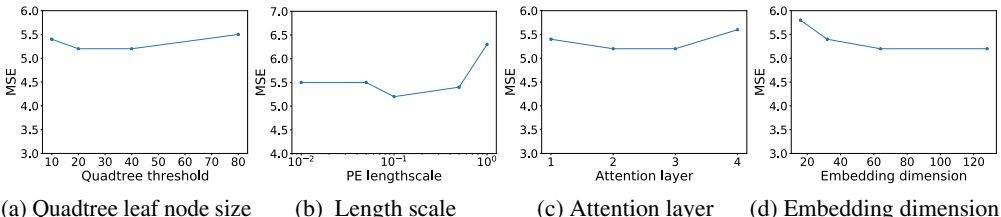

(a) Quadtree leaf node size  (b) Length scale  (c) Attention layer  (d) Embedding dimension

Figure 7: Parameters sensitivity analysis.

We also conducted a sensitivity analysis of our model to different hyper-parameters, including the quadtree leaf node size threshold $M$, spatial position encoding (kernel bandwidth) length scale $\sigma$, the number of attention layers in the encoder and decoder, and the embedding dimension of the latent representation. We use the red tide dataset as an example to conduct the sensitivity analysis. When evaluating the influence of one parameter, we keep all other parameters unchanged. The results are summarized in Figure 7. First, we evaluate model sensitivity to the threshold of quadtree leaf node size. We can see that the MSE first decreases with a $20$ or $40$ threshold because the neighbor samples increase. The MSE then increases when increasing the threshold to $80$, because the quadtree depth is too small, and less hierarchical information is contained in the tree. Second, we evaluate model sensitivity to the length scale of spatial position encoding, which determines the model's initial auto-correlation range in the continuous space. We change the length scale of spatial position encoding from 0.01 to 1. The corresponding test MSE scores are in Figure 7(b). The model MSE first drops to $0.51$ and then increases to $6.2$. The model is generally stable in the length scale range of $0.01$ to $0.5$. Then we change the number of attention layers in the model from 1 to 4. The results in Figure 7(c) show that the performance is best when using 2 spatial attention layers, and the performance is stable. Then we analyze the influence of latent representation dimension in Figure 7(d). We change the embedding dimension from 16 to $32, 64, 128$, and it shows that embedding dimension 64 performs well and the model performs stable if continuing to increase the latent dimension.

## 7.4 Uncertainty Quantification

We provide additional uncertainty quantification results for the turbidity dataset in Table 5. We can observe that our method outperforms existing uncertainty quantification baseline methods.

Table 5: Comparison on uncertainty quantification performance on turbidity dataset

| Model | Accurace | Uncertainty | | AvUa/ AvUc | AvU |
|---|---|---|---|---|---|
| GP | | Certain | Uncertain | | 0.23 |
| | Accurate | 254 | 1548 | 0.14 | |
| | Inaccurate | 343 | 617 | 0.65 | |
| Deep GP | | Certain | Uncertain | | 0.38 |
| | Accurate | 436 | 1466 | 0.23 | |
| | Inaccurate | 315 | 545 | 0.64 | |
| Spatial GNN | | Certain | Uncertain | | 0.44 |
| | Accurate | 1238 | 730 | 0.63 | |
| | Inaccurate | 526 | 268 | 0.34 | |
| Galerkin Transformer | | Certain | Uncertain | | 0.40 |
| | Accurate | 1036 | 872 | 0.54 | |
| | Inaccurate | 602 | 252 | 0.30 | |
| HST (Our model) | | Certain | Uncertain | | 0.49 |
| | Accurate | 828 | 1194 | 0.41 | |
| | Inaccurate | 297 | 443 | 0.60 | |