# OpenReview forum: "A Hierarchical Spatial Transformer for Massive Point Samples  in Continuous Space"
_NeurIPS.cc/2023/Conference — NeurIPS 2023 poster_

### Official Review · Reviewer_Y8FJ · 2023-07-06

**Soundness:** 4 excellent
**Presentation:** 3 good
**Contribution:** 4 excellent
**Rating:** 8
**Confidence:** 4

**Summary:**

The authors proposed a hierarchical spatial transformer model for many irregular point samples in continuous spatial domain. Compared with existing methods, the proposed method can model implicit spatial dependency across irregular samples and in multiple scales in continuous space. The proposed model uses a quad-tree hierarchy to conduct efficient attention operations by approximating distant points in a coarse resolution. The model also includes an uncertainty quantification module to capture the varying prediction confidence in different input sample density. Evaluations on several datasets confirm that our method outperforms multiple baselines in accuracy and effectiveness of uncertainty quantification.

**Strengths:**

Overall, this is a solid paper with novel technical contributions and extensive experimental evaluations.

• The paper solves a significant problem of spatial representation learning for irregular samples in continuous space. The problem has many significant applications in environment sustainability. It is also important for learning surrogate models to speed up numerical simulations.

• The technical novelty of the proposed hierarchical attention architecture is strong. The model uses a quad-tree to learn latent representation of point samples in different subareas in a multi-scale hierarchy. The attention layers use quad-tree to make spatial approximation to overcome the computational bottleneck. The model also has an uncertainty quantification module.

• The proposed method has shown promising results on both real-world and synthetic datasets. The evaluation is solid with sensitivity analysis and computational experiments.

**Weaknesses:**

Although the experimental evaluation is extensive, it will be helpful to add some more on how the thresholds of uncertain/certain predictions are determined in UQ metric.

**Questions:**

• How are the thresholds of uncertain/certain predictions determined in UQ metric?

**Limitations:**

No limitations on societal impacts.

---

> ### Author Rebuttal · Authors · 2023-08-10
>
> Question: how are the thresholds of uncertain/certain predictions determined in UQ metric?
>
> Response: Thank you for your positive feedback. About how to choose thresholds of uncertainty in the UQ metric, we discussed them in the supplementary materials in section 7.1. Here we briefly describe how we choose the UQ threshold. We used a validation dataset for determining accurate/inaccurate and certain/uncertain thresholds. Specifically, the threshold for accuracy is determined by the average MAE loss on validated samples. Then the threshold for uncertainty is determined by the average uncertainty score for accurate and inaccurate prediction respectively. The detailed equation is provided in our supplementary materials section 7.1.

---

> > ### Comment · Reviewer_Y8FJ · 2023-08-18
> > **Thank you for the response**
> >
> > Thanks for the responses. The authors have properly addressed my questions. I am fine with an acceptance.

---

### Official Review · Reviewer_J8MW · 2023-07-06

**Soundness:** 3 good
**Presentation:** 3 good
**Contribution:** 3 good
**Rating:** 4
**Confidence:** 4

**Summary:**

This paper proposes a hierarchical transformer to model a large number of irregular point samples in continuous space. This is achieved by a quad-tree hierarchy which could learn the multi-scale spatial representation. So the long-range interactions are recorded.

The experiment is performed in three real-world dataset (two water quality datasets and one sea-surface temperature prediction) and one simulation dataset.

**Strengths:**

This submission tries to solve the large number of issues by using quadtree. In my understanding, the quadtree is a classic technique for downsampling points so that the point number will be reduced to an affordable level.  And the quadtree could model the long-range interactions without the distribution assumption.

The proposed method reduced the computational cost to $O(N\log N)$ or $O(NM)$.

The approach presentation is detailed and clear.

**Weaknesses:**

The adopted quadtree technique seems could only work well in the sparse case. As a result, the proposed method is not a general solution for all kinds of large numbers of point samples.

The experiment setting is not convincing in the reviewer's current understanding. In particular, Red Tide and Turbidity datasets have a large number of data samples, but it seems only to have one 'epoch'. In other words, The point number is large but the total data size seems very small. And for Darcy flow and Sea Surface, it seems each image only have 400 point samples, it is not a large number set in my understanding, the all-pair transformer could cover it.

**Questions:**

Could the author clarify how the adopted quadtree technique in the paper could solve uniformly distributed cases?

Link to the weakness above, as Darcy flow and Sea Surface datasets are set, could the author claim why do we have to feed all point samples in a single feed? In other words, why don't we solve this task in all three datasets by following a classic 'classification setting'?

Compared with the all-pair transformer, pros and cons?

Could the reviewer know the number of parameters for all-pair transformer and other baselines in Table 2?

Any guidance in choosing hyper-parameters for designing quadtree if needed?

As in Weakness, my main concern is first two questions.

---

> ### Author Rebuttal · Authors · 2023-08-10
>
> Dear reviewer:
>
> **W1 & Q1: how the quadtree technique could solve uniformly distributed cases.**
>
> Thank you for the comment. In fact, our HST model does not require samples to be sparsely distributed for efficiency gain. For uniformly distributed point samples, the quadtree will be a balanced (complete) tree.  We can illustrate how our model works in this case with an example. Please see the example in Fig.1 of global rebuttal pdf (the topmost box). There are 8*8 point samples evenly distributed in 2D. Assume the minimum leaf node size is 4. The height of the quadtree is $h =log_4(64/4) = 2$, as Fig.1(b) shows. Consider the query sample of point 1, the key node set include 3 remaining points in the same leaf node (points 2-4), 3 internal nodes $(r_2^2 ... r_4^2)$, 3 internal nodes $(r_2^1 ... r_4^1)$. Thus, there are only 9 keys, much lower than the total number of points (64). In fact, our efficiency gain is better for uniformly distributed case, as the depth of the tree is lower. This significantly reduces the time costs.
>
> **W2 & Q2: the experiment with 400 point samples.**
>
> This is a valid concern. First, we chose to randomly select 400 input points from each $64\times 64$ image in the Darcy flow and Sea Surface dataset because drawing much denser input samples (e.g., over 1000) makes the problem trivial because of strong spatial autocorrelation. In this case, even a simple nearest-neighbor estimator can work well.  More importantly, even 400-point input samples is not as easy as it seems for the vanilla transformer model due to the high memory and time costs. The GPU memory costs for vanilla all-pair transformer with 1K point samples exceed the capability of A100 GPU due to the large size of intermediate tensors. This is confirmed in our experiment results (Figure 4 in the paper).
>
> For detailed calculation, please see our calculation below.  Second, we can run all the point samples in a single feed, but the GPU memory cost is very high. We provide a memory consumption analysis below. Third, even with 400 point samples, our model can still improve the computational efficiency as Fig.4 in our paper shows. The computation time is reduced from 25 minutes to 5 minutes for one epoch. Our model can have slight improvement compared to all-pair transformer as Table 2 shows.
>
> Memory cost estimation of vanilla transformer:
>
> Assume $L = 1,000$ points, $H = 8$ attention heads, $N = 6$ block, the dimension of each head $D = 64$, and the batch size $B= 512$. The model stores results from forward attention activation (including self-attention, dense layer) ($M_{a}$) and backward gradient ($M_{g}$), and model parameters ($M_{m}$). The dominant factor is the attention layer. For each attention layer, given query and key tensors $Q,K \in R^{L*D}$, the attention matrix multiplication will result in a tensor with the shape of $[L, L]$ for each attention head and each sample within the minibatch. Thus, the total memory cost for all heads in one minibatch in one attention layer is: $ M_{a} = BHL^{2} \times (float32 \ bytes)$. Multiplying this by $N$ attention layers, the cost is $4NBHL^{2} \ bytes= 6\times 512\times 8\times 4\times 1000^{2} bytes \approx  94 GB$. This exceeds A100 memory capacity (80GB), as shown in our experiment results in Figure 4.
> In contrast, our HST model significantly reduces the memory costs to only 24 GB.  The reason is that we approximate point samples far away from the query points by coarse representation (in quadtree internal nodes) in the attention computation. The further away points are from the query points, the coarser cells we use in approximation.  Due to limited space, we do not provide  estimation here. The detailed derivation can be found in the rebuttal for reviewer 2.
>
> **Q3: pros and cons compared with all-pair transformer.**
>
> This is a great question. We provide some analysis on the pros and cons of our model compared to the all-pair transformer.
> The pros:
> 1.	Able to capture multi-scale effect with the hierarchical quad-tree structure: Our model learns a hierarchical spatial representation within the quadtree. The multi-scale effect is important for many scientific problems. This is reflected by the improved prediction accuracy in experiments.
> 2.	Our HST reduces the time and memory costs when compared with the all-pair transformer. Please see our analysis above.
> 3.	We added uncertainty quantification branch to estimate the model’s confidence.
>
> The con:
> 1.	Our HST makes a trade-off between efficiency and spatial granularity when calculating attention between points. We approximate the set of key points by coarse cells in the quadtree based on distance to the current query point.
>
> **Q4: the number of parameters of baselines:**
>
> Thanks for the great suggestion. We will add the number of parameters for the baselines and our model in Table 2. The numbers are as follows.  GP: 2 (RBF kernel parameters), Deep GP: 0.24 million, Spatial GCN: 0.79 million, All-pair transformer: 3.81 million, HST model: 3.81 million, NodeFormer: 2.2 million, Galerkin Transformer: 17.6 million
> Note that the large number of parameters in Galerkin Transformer is because we used the default code setup without tunning. We could set up the number of attention layers the same as other transformer models (six encoder-decoder layers). In this case, the number of Galerkin transformer parameters would be 4.4 million.
>
> **Q5. the guidance on choosing hyper-parameters for designing quadtree.**
>
> We provide the quadtree hyperparameters in our supplementary material Section 7.1 and the sensitivity analysis. We chose the minimum leaf node size of 20 in experiments based on prior knowledge about the spatial dependency structure. The size of the leaf node can be determined based on the spatial homogeneity and efficiency needs. Roughly, the more spatially homogeneous the input point samples are and the more efficiency gain we need, the bigger leaf node cells we need to choose.

---

### Official Review · Reviewer_Ma6V · 2023-07-07

**Soundness:** 3 good
**Presentation:** 3 good
**Contribution:** 3 good
**Rating:** 5
**Confidence:** 5

**Summary:**

This paper proposes a quad-tree partition of irregularly distributed sample locations. This leads to an algorithm with O(NlogN) complexity.

**Strengths:**

Quad-tree idea is nice, even though not original for irregular grids or pixels.

**Weaknesses:**

The targeted problem in the paper is regression. But the model is framed as an encoder-decoder. The paper does not explain why this seemingly unreasonable choice.

The O(NlogN) complexity has a large big-O constant because of sparse matrix operations which are known to be inefficient on GPUs.

There are not enough details on models and experiments. For example, the model parameters such as number of layers, hidden size and number of attention heads are not clearly specified. The training hyper parameters such as learning rate and its decay schedule are not explicitly stated. This makes it hard to determine the quality of the experiments, let alone replicating them.

Datasets are all 2D with up to 1K sample points. These are well within the capacity of vanilla Transformer on A100, the GPU used in the paper. And the optimal leaf node size is 100. This results in a very shallow quad tree. So the evidence about the efficiency gain is not convincing.

**Questions:**

What is the vocab for the decoder? The solution space for regression is not discrete.

---

> ### Author Rebuttal · Authors · 2023-08-10
>
> Dear reviewer:
>
> **W1 & Q1:the encoder-decoder module for regression and the vocab for the decoder.**
>
> Response: The confusion may come from the naming. Although the encoder-decoder architecture was originally used in machine translation (discrete vocab), common encoder-decoder architectures, e.g., U-Net, transformer, have been widely used in regression problems such as temperature and precipitation prediction [1-4]. In a general sense, an encoder learns a latent feature representation that encodes the structural dependency within input data. A decoder predicts the target variable from encoded features. There is no restriction that the framework cannot be applied to regression. We hope this can address the concern and please comment if any questions.
>
> [1] Autoformer: Decomposition Transformers with Auto-Correlation for Long-Term Series Forecasting. NeurIPS 2021
>
> [2] Informer: Beyond Efficient Transformer for Long Sequence Time-Series Forecas. AAAI 2021
>
> [3] Spatiotemporal Swin-Transformer Network for Short Time Weather Forecasting. CIKM 2021
>
> [4] NumHTML: Numeric-Oriented Hierarchical Transformer Model for Multi-Task Financial Forecasting. AAAI 2022
>
> **W2: the O(NlogN) complexity has a large big-O constant**
>
> Response: We agree that sparse matrix operations can be less efficient in GPU penalization when compared to normal dense matrix operations. However, there are already significant efforts in GPU optimization of sparse matrix multiplication (e.g., cuSPARSEt library). In our HST model, the large big-O constant is not a dominant factor, as shown in experiment results in Figure 4. For 400 points, our HST reduces the time costs from 25 mins to 5 mins. One note is that the time cost for the default all-pair transformer when the input size is 1000-point is not measurable because out-of-memory problem (please see our response below for this question). In contrast, our model can run 1000 points within twenty minutes for one epoch.
>
> **W3: not enough details on models and experiments.**
>
> Response: In fact, we had provided all the model hyper-parameters and training details in our supplementary material section 7.1. The description is quoted as following:
> Our HST model used six spatial attention layers in both the encoder and the decoder. We use 8 heads and the latent representation embedding dimension of each head was 64, and the quadtree leaf node size threshold was 20 by default. The batch size was $512$. For the training process, we used the MSE loss with a decaying learning rate that reduced the learning rate by half if the validation loss did not improve over five epochs (with an initial
> learning rate of $10^{-4}$ and a minimum rate of $10^{-7}$). We also used early stopping with a patience of 10 epochs and a maximum of 50 epochs. The optimizer was Adam with $\beta_1=0.9$ and $\beta_2 = 0.98$. The $L_2$ regularization weight was $10^{-4}$.
>
>
> **W4: 1K samples within the capability of vanilla transformer**
>
> Response:
> This is a good question. First, we want to clarify that even 1K points will exceed the capacity of vanilla (all-pair) Transformer on A100 GPU. This is because the dominating GPU memory costs come from intermediate tensors in forward propagation. Please see an example below on the calculation of GPU memory cost with 1K point samples for both vanilla (all-pair) transformer and our HST transformer.
>
> Memory cost estimation of vanilla transformer:
>
> Assume $L = 1,000$ points, $H = 8$ attention heads, $N = 6$ block, the dimension of each head $D = 64$, and the batch size $B= 512$. The model stores results from forward attention activation (including self-attention, dense layer, and skip connection) ($M_{a}$) and backward gradient ($M_{g}$), and model parameters ($M_{m}$). The dominant factor is the attention layer. For each attention layer, given query and key tensors $Q,K \in R^{L*D}$, the attention matrix from $QK$ multiplication will result in a tensor with the shape of $[L, L]$ for each attention head and each sample within the minibatch. Thus, the total memory cost for all heads in one minibatch in one attention layer is: $ M_{a} = BHL^{2} \times (float32 \ bytes)$. Multiplying this by $N$ attention layers, the cost is $4NBHL^{2} \ bytes= 6\times 512\times 8\times 4\times 1000^{2} bytes \approx  94 GB$. This exceeds A100 memory capacity (80GB), as shown in our results in Figure 4.
> In contrast, our HST model significantly reduces the memory costs.  The reason is that we approximate point samples far away from the query points by coarse representation (in quadtree internal nodes) in the attention computation. The further away points are from the query points, the coarser cells we use.
>
> HST memory cost estimation:
>
> Assume same setup ($L = 1K$ points, $H = 8$ attention heads, $N = 6$ block, the dimension of each head $D = 64$, and the batch size $B= 512$) and a quadtree height $h=10$ and minimum leaf node size $T=100$. HST has the same number of query points. For each query point in a leaf node, the key set includes all the points in the same leaf node, as well as only a subset of internal quadtree nodes (the leaf’s siblings, its parent’s siblings, and its grandparent’s siblings, …, till the root node). This is illustrated in the red boxes in Fig.1 in global rebuttal box. In this example, the key set is reduced from 64 to 9 nodes. In general, the key set is reduced from $L$ to $T+3h$. Thus, the cost of $N$ attention layers is reduced from $NHL^{2}$ to $NHL(T+3h)$. The sparse indicator matrix for selective keyset attention calculation in HST has a shape of $\mathbf{I} \in R^{L \times 2L}$, and its non-zero element count is $L(T+3h)$, thus the memory cost is $BNL(T+3h)\times float32 \ bytes $. Thus, the total memory cost is $2BNL(T+3h)\times float32 \ bytes
> = 2\times 512\times 6\times 8\times 1000\times 130\times 4 \ bytes \approx 24 GB$. Similarly, the time costs of HST are also lower than the vanilla transformer due to less attention pairs. See our responses above.

---

> ### Comment · Reviewer_Ma6V · 2023-08-20
>
> Thanks for the clarification. I have increased my ratings.

---

### Official Review · Reviewer_7rvT · 2023-07-07

**Soundness:** 4 excellent
**Presentation:** 3 good
**Contribution:** 4 excellent
**Rating:** 8
**Confidence:** 4

**Summary:**

This paper proposes a hierarchical spatial transformer model for a large number of point samples in continuous space. The model is important for geoscience applications, such as water quality monitoring and air quality monitoring, and operator learning for numerical models. The novel idea includes continuous position-encoding and hierarchical attention layers, which make a trade-off between efficiency and spatial resolution and captures the interactions in multiple spatial scales. The proposed model also has uncertainty quantification that reflect the effect of sample spatial sparsity. The proposed method is compared against multiple baselines on different datasets with significant improvements. There are also computational experiments to show the efficiency of the method.

**Strengths:**

1. The idea of hierarchical attention with a quad-tree structure for massive samples in the continuous space is quite novel. The idea makes a trade-off between computational efficiency and spatial approximation of latent representation. The spatial approximation in a quad-tree structure is well-motivated by the spatial autocorrelation effect.

2. The proposed method has strong technical contributions. There is theoretical analysis of the time complexity of the proposed model and the gain in efficiency.

3. There are extensive experimental evaluations against multiple state of the art methods on both real-world and synthetic datasets. The results show the better accuracy and uncertainty quantification of the proposed method.

**Weaknesses:**

1. There are some minor presentation issues that can be fixed. For example, Table 2 has a typo in the title. It should be “two real-world datasets and a synthetic dataset”.
2. The application paragraph in the introduction can be more detailed to better highlight the impact of the model in operator learning for numerical simulation.

**Questions:**

1. Can you explain in more details how the proposed model can be used in numerical simulations (e.g., ocean current, multiphase flow) in the introduction?

**Limitations:**

No limitations discussed in the paper.

---

> ### Author Rebuttal · Authors · 2023-08-10
>
> Dear reviewer:
>
> Thank you for your positive feedback and careful reading. We have done thorough proofreading and corrected the typos in the paper. For the second question about how the proposed model can be used in numerical simulations, we use multiphase flow as an example to explain. In this case, the input features consist of 3D coordinates of point samples in the continuous space as well as the initial pressure and velocity of those samples, the target output is the acceleration at any point location. We can add this into the paper.

---

> > ### Comment · Reviewer_7rvT · 2023-08-15
> > **Rebuttal read**
> >
> > Thank you for providing the rebuttal. I believe they addressed my previous questions. The paper makes valid and practical contributions to the machine learning community. The proposed idea can also beneifit a broad range of scientific domains. In my opinion the paper should be accepted.

---

### Author Rebuttal · Authors · 2023-08-10

This PDF contains a figure illustration of evenly distributed input points. It illustrates how our model can reduce computational costs in this case by selecting a subset of keys in attention calculation.

---

### Comment · Area_Chair_Jy8N · 2023-08-18

Dear authors, thank you for your rebuttal. I will take into consideration your unanswered responses about the computational complexity, applicability of the method, and comparison with other transformers.

---

### Decision · Program_Chairs · 2023-09-21

**Decision:**

Accept (poster)

**Comment:**

This paper introduces a transformer architecture for spatial modeling problems with irregularly spaced sample points. The authors use a quad tree structure to reduce the computational requirements of the transformer, and introduce a decoder mechanism to quantify uncertainty. Overall, the proposed method, which contains many novel contributions, can be applied to many important application domains. A few concerns about the method include: (1) it will only be effective in certain settings (irregularly spaced points, not too many points), (2) it will have high computational costs in these other settings, and (3) the sparse operations cannot be significantly accelerated by GPUs. That being said, the efficacy of the proposed method, as well as the novel approach of combining quad trees with transformer modules, will be of great interest to the NeurIPS community and therefore I recommend acceptance.